# Evaluating the influential factors for life preserver donning tests

**Ruiliang Yang[1], Zijiang Wu[2], Xiaoming Qian[2]\***

**1** Key Laboratory of Modern Electromechanical Equipment Technology, Tiangong University, Tianjin, China,
**2** School of Textile Science and Engineering, Tiangong University, Tianjin, China

\* qxm@tiangong.edu.cn

**Data Availability Statement:** All relevant data are within the paper and its Supporting information files.

**Funding:** This study was supported by the National Natural Science Foundation of China (No. U1933111).

## Abstract

Life preservers often play a vital role in ensuring passenger safety in water-related accidents, while the difficulty of donning life preservers has been repeatedly proved even in a donning test. To evaluate the influencing factors for life preserver donning tests, 109 college students and 42 villagers were chosen as subjects. A total of fourteen variables with seven categorical variables and seven continuous variables were considered as potential influencing factors. T-test and one-way analysis of variance (ANOVA, for three or more categories) were used to judge whether grouping in categorical variables had a significant effect on the donning performance. Then all variables were offered into the stepwise linear regression (SLR) to evaluate the influential factors for life preserver donning tests. Results showed that four of fourteen variables, including gender, instruction condition, age group, and tool test time (representing the subject's flexibility), had a significant effect on the donning performance. To evaluate the relationship between the donning performance and influencing factors, models of the retrieving time, the opening time, and the donning time were built based on the SLR analysis. The paper also highlights recommendations for modification of the donning test procedure, which helps to improve the validation and reliability of life preserver donning tests.

## Introduction

Life preservers often play a vital role in ensuring passenger safety in water-related accidents and are required on an airplane in overwater operations by the regulations of many countries [1]. However, the difficulty of donning life preservers had been repeatedly proven by accident reports [2], research papers [3–5], and donning tests [6–8]. In the accident ditching on the Hudson River in 2009 [2], only 4 out of 150 passengers were able to correctly don their life preservers, which highlighted the unreliability of life preservers. To evaluate donning performance of the life preserver, Corbett et al. [6] tested typical life preservers in 2014. Results showed none of life preservers met the donning requirement of the Technical Standard Order (TSO) series standard. This finding was consistent with two other Federal Aviation Administration (FAA) reports [6,7]. Considering this fact that none of life preservers can reach the mandatory donning requirement, why are there still many life preservers on the market? The

**Competing interests:** The authors have declared that no competing interests exist.

main reason is that donning tests of life preservers do not always yield reliable results, so the regulatory authorities and manufacturers often ignore the mandatory donning requirement of the life preserver [1]. Thus, it is necessary to research the influencing factors to improve the validation and reliability of donning tests. This paper aims to evaluate the influential factors for donning tests of adult life preservers, and propose recommendations for modification of the donning test procedure.

TSO-C13g [9] formulated by the Department of Transportation of the US specified the minimum performance of adult life preservers. It required: "at least 75% of the total number of test subjects, and at least 60% of the test subjects in each age group, can don the life preserver within 25 seconds" in the donning test, and "75% of the total number of test participants must complete package in less than 7 seconds" in the package opening test.

The influential factors related to the donning test in TSO-C13g [9] include age, gender, height, weight, and head circumference. The age requirement in TSO-C13g [9] was that test subjects should be distributed at least five groups: 20–29, 30–39, 40–49, 50–59, 60–69 years. Any age group may not exceed 30% of the total number. The gender requirement in TSO-C13g [9] was that the same sex should not exceed 60% of the total number of test subjects in the donning test. As for the package opening test, the package should be opened less than 7 seconds by at least 8 of 10 females over the age of 60, or within 10 seconds by 8 of 10 females with reduced dexterity. The subject characteristics defined by TSO-C13g [9] are shown in Table 1.

## Method

### Ethics statement

The study was approved by Human Research Ethics Committee for Non-Clinical Faculties of School of Mechanical Engineering, Tiangong University. All participants were verbally informed of the contents of the experiment, then signed their names to agree the experiment. The participants in the figures have given written informed consent to publish their images.

### Experimental design

The research was conducted in a laboratory of Tiangong University. The laboratory was divided into two areas: an air carrier coach platform area (see Fig 1) and a preparation area (see Fig 2).

The air carrier coach platform consisted of two rows of air carrier coach class triple-seat, an experimental monitoring system, and a simulated cabin floor. The air carrier coach was purchased from commercial airlines. Seats, seat belts, life preservers, and armrests were all in good condition, fully meeting the test requirements. The experimental monitoring system was composed of two surveillance cameras, which can monitor the test process from the front and back directions. The simulated cabin floor was made of 15mm high steel plate to simulate the floor of the commercial aircraft cabin.

**Table 1. Test subject characteristics.**

|  | 5th percentile | | | 50th percentile | | 95th percentile | | |
|---|---|---|---|---|---|---|---|---|
|  | Height (m) | Weight (kg) | Head circumference (cm) | Height (m) | Weight (kg) | Height (m) | Weight (kg) | Head circumference (cm) |
| Male | ≤1.7 | ≤63.9 | Nil | 1.8 | 79.4~85.6 | ≥1.9 | ≥110.7 | ≥60.4 |
| Female | ≤1.5 | ≤51.4 | ≤52.5 | 1.6 | 64.4~70.7 | ≥1.7 | ≥93.0 | Nil |

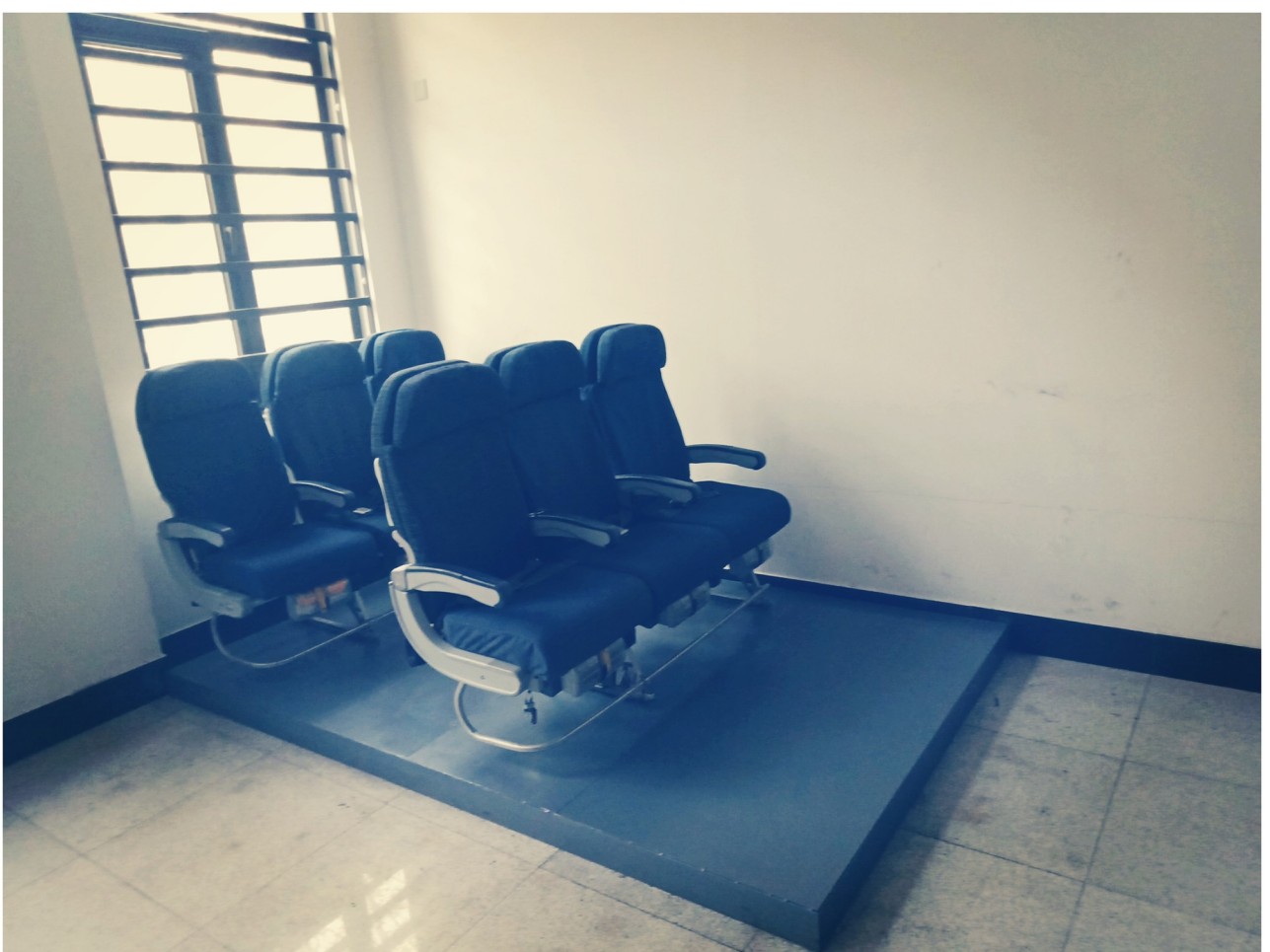

**Fig 1. The air carrier coach platform area.**

The preparation area was utilized to record demographic information, fill in the questionnaire, and carry out the General Aptitude Test Battery (GATB) [10]. Demographic information included name, gender, age, height, urban/rural, villager/student, and wearing glasses or not. Measurement data included weight, body fat rate, and head circumference. Weight and body fat rate of test subjects were measured by Mi Body Composition Scale 2, and head circumference was measured by laboratory staffs.

The GATB is a test compiled by the Employment Insurance Bureau of the US Department of Labor, which has excellent reliability and validity to evaluate individual flexibility [10]. To more effectively describe human flexibility, the GATB used in this paper included self-test questions and tool tests. Self-test questions consisted of nine items (general intelligence, verbal ability, numerical aptitude, spatial relation, shape perception ability, clerical awareness, motor coordination ability, finger flexibility, and wrist flexibility). Each item was composed of five questions graded on a five-level scale (1, strong; 2, just strong; 3, average; 4, just weak; 5, weak). The total score of these nine items was regarded as a self-test score, and the max self-test score was 225. Tool tests included placing tool test, turning tool test, assembling tool test, and disassembling tool test. These tests required the test subjects can complete placing task, turning task, assembling task, and disassembling task as quickly as possible, and the total time

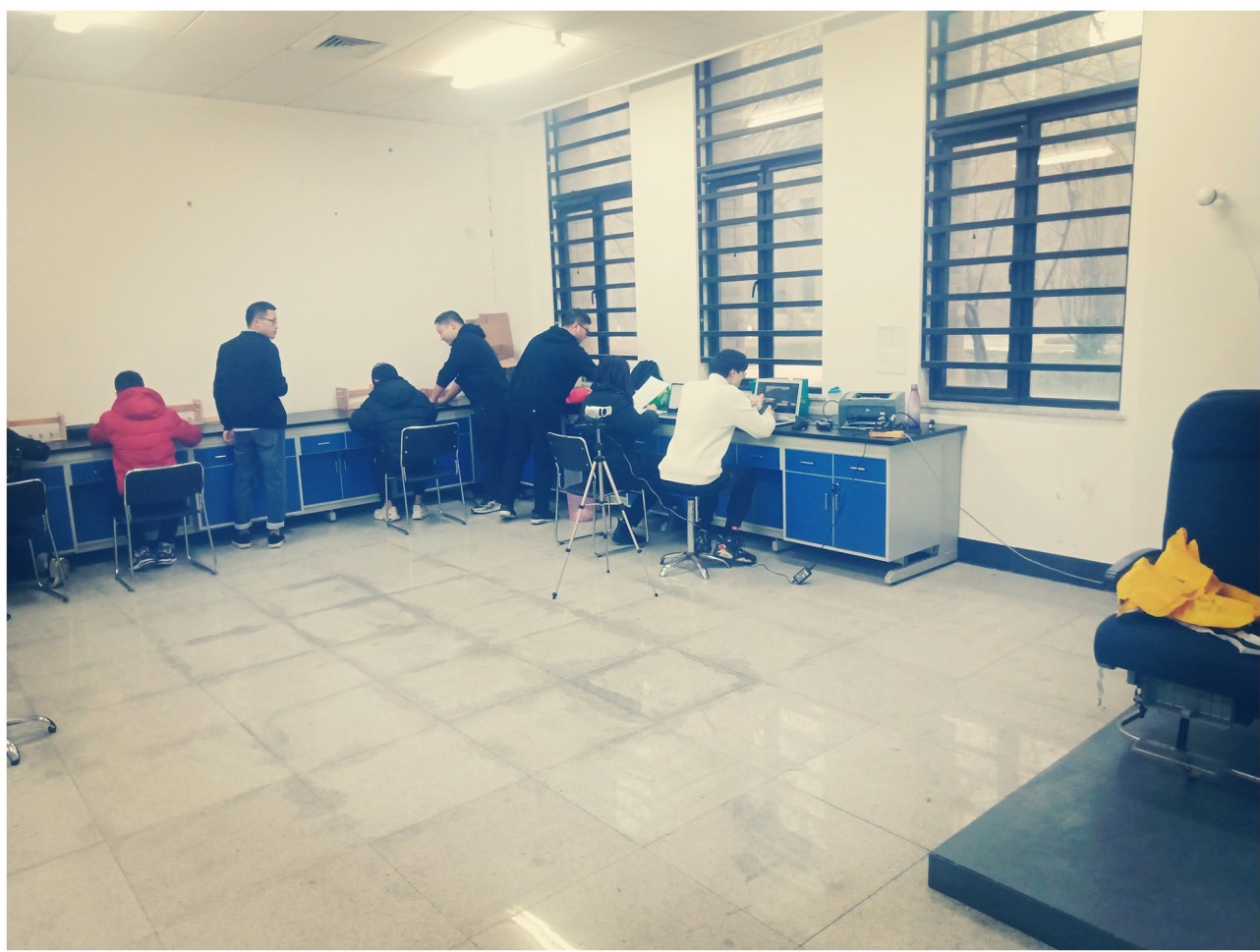

**Fig 2. The preparation area.**

of these four tasks was regarded as the tool test time. Lower self-test score and shorter tool test time meant participants were more flexible.

## Participants and life preservers for test

A total of 151 subjects were recruited in the test, including 85 males and 66 females. Among the test subjects, 109 were undergraduates of Tiangong University and 42 were villagers near Tiangong University. None of the subjects had any knowledge of life preservers before this test.

Life preservers for the test were typical inflatable aviation life preservers used by major airlines at present. Each life preserver was composed of upper and lower chambers, straps, inflation gas reservoirs, oral inflation means, and survivor locator light.

## Procedure

1. Three or two subjects were in one group and participated together in the donning test. After entering the laboratory, the subjects should fill in the information collection form.

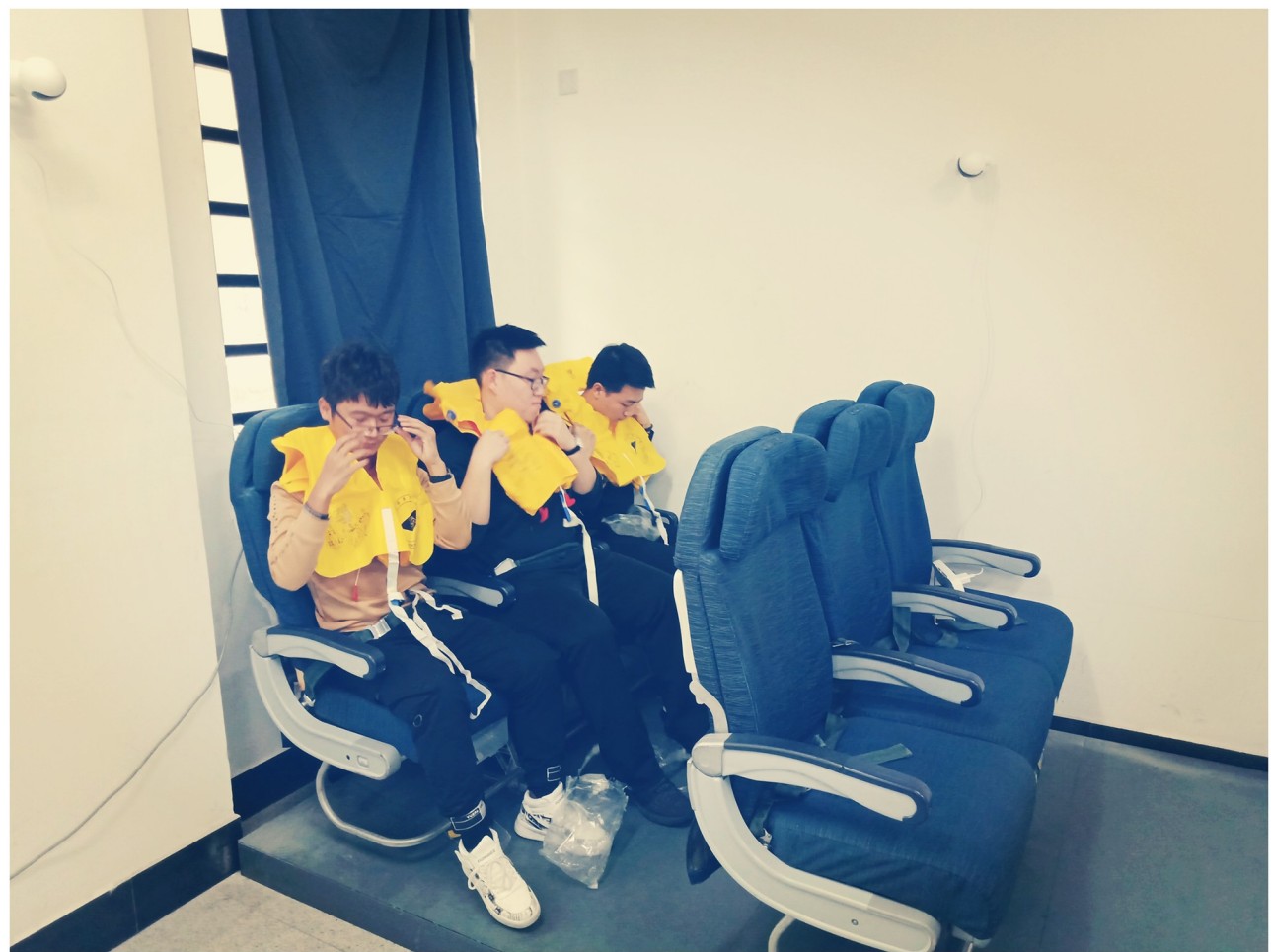

**Fig 3. Life preserver donning tests.**

Then weight, head circumference, and body fat rate of the subjects were measured and recorded by laboratory staff, followed by self-test questions and tool tests.

2. Test subjects sat side by side in the second row of air carrier coach and fastened their seat belts. Two levels of instruction condition were used before the test. Instruction condition I was subject's reading briefing card and staff's oral briefing. Instruction condition II was subject's reading briefing card and staff's donning demonstration. After donning instruction was conducted, the donning test (see Fig 3) began.

3. When the test subject retrieved the life preserver from under the seat, the test started timing. The subject opened the package and began to don it. When the test subject completed fastening and adjusting the life preserver straps, the test signaled to end the test.

4. Overall TSO test time was composed of the retrieving time, the package opening time, and the donning time. The three types of time were divided according to the video after the test. Two time-division points were when the package was higher than the knee and when the life preserver was taken out of the package, respectively.

**Table 2. Potential categorical influential factors for life preserver donning tests.**

| Variable | Potential categorical influential factor | Category | Frequency | Percent (%) |
|---|---|---|---|---|
| $x_1$ | Gender | Male | 85 | 56.3 |
| | | Female | 66 | 43.7 |
| $x_2$ | Urban/rural | Urban | 62 | 41.1 |
| | | Rural | 89 | 58.9 |
| $x_3$ | Villager/student | Villager | 42 | 27.8 |
| | | Student | 109 | 72.2 |
| $x_4$ | Wearing glasses | Yes | 76 | 50.3 |
| | | No | 75 | 49.7 |
| $x_5$ | Seat | Aisle seat | 51 | 33.8 |
| | | Middle seat | 51 | 33.8 |
| | | Window seat | 49 | 32.4 |
| $x_6$ | Age group | <20 | 80 | 53.0 |
| | | 20–29 | 30 | 19.9 |
| | | 30–39 | 5 | 3.3 |
| | | 40–49 | 25 | 16.6 |
| | | 50–59 | 11 | 7.3 |
| $x_7$ | Instruction condition | Instruction condition I | 122 | 80.8 |
| | | Instruction condition II | 29 | 19.2 |

## The potential influential factors for life preserver donning tests

A total of fourteen variables with seven categorical variables and seven continuous variables were considered as potential influencing factors. Seven categorical variables include gender, urban/rural, villager/student, wearing glasses, instruction condition, seat, and age group, see Table 2. Seven continuous variables include height, weight, head circumference, body fat rate, metabolic rate, self-test score, and tool test time, see Table 3. Through data consolidation, Tables 2 and 3 summarizes the demographic information of all participants. About 56.3% of the participants were male, and 43.7% were female. Test subjects included 72.2% of college students and 27.8% of villagers. Aisle seat, middle seat, and window seat all accounted for about 1/3. According to TSO-C13g [9], five age groups used in this paper were <20, 20–29, 30–39, 40–49, and 50–59 years. About 53.0% of the participants were under the age of 20 years old. There were two levels of instruction condition. Instruction condition I was to let test subjects read the briefing card, then the staff orally briefed the donning process. Instruction condition

**Table 3. Potential continuous influential factors for life preserver donning tests.**

| Variable | Potential continuous influential factor | Mean | Std. | Min. | 25th percentile | Median | 75th percentile | Max. |
|---|---|---|---|---|---|---|---|---|
| $x_8$ | Height (cm) | 169.2 | 8.7 | 152 | 161 | 170 | 177 | 192 |
| $x_9$ | Weight (cm) | 66.8 | 14.9 | 44.3 | 55.4 | 65.2 | 73.6 | 120.1 |
| $x_{10}$ | Head circumference (cm) | 56.8 | 2.2 | 50.5 | 55.2 | 56.7 | 58.2 | 67.6 |
| $x_{11}$ | Body fat rate (%) | 24.5 | 9.8 | 5 | 16.7 | 25.4 | 31.8 | 43.5 |
| $x_{12}$ | Metabolic rate (w/m$^2$) | 1445.4 | 266 | 1005 | 1236.3 | 1393 | 1608 | 2377 |
| $x_{13}$ | Self-test score | 126.7 | 20.7 | 69 | 114 | 126 | 138 | 191 |
| $x_{14}$ | Tool test time (s) | 247.4 | 45.9 | 163 | 213.3 | 238 | 272.5 | 389 |

**Table 4. The pass percentage of different genders, different subjects, and different instruction conditions.**

|  | Male student | Female student | Male villager | Female villager | Instruction condition I | Instruction condition II | Total |
|---|---|---|---|---|---|---|---|
| Number | 67 | 42 | 18 | 24 | 122 | 29 | 151 |
| Package opening time ≤ 7s, n (%) | 32 (47.8%) | 12 (28.6%) | 11 (61.1%) | 7 (29.2%) | 48 (39.3%) | 14 (48.3%) | 62 (41.1%) |
| Donning time ≤ 25s, n (%) | 18 (26.9%) | 7 (16.7%) | 6 (33.3%) | 1 (4.2%) | 14 (11.5%) | 18 (62.1%) | 32 (21.2%) |

II was to let test subjects read the briefing card, then the staff demonstrated how to don the life preserver.

## Statistical analysis

All data analyses were performed using SPSS (v22) software. To judge whether grouping in categorical variables had a significant effect on the donning performance, T-test was used for two-level categorical variables such as gender, urban/rural, villager/student, wearing glasses, and instruction condition. One-way analysis of variance (ANOVA) was adopted for categorical variables with three or more categories such as seat and age group. All variables were offered into the stepwise linear regression (SLR) [11] to avoid some of common problems associated. In the stepwise linear regression models, dummy variables were created for the categorical variables, using the first or lowest category as the reference category. Pearson correlation analysis [12] was also used to assess the strength of the relationship between dependent and independent variables. All tests were conducted at a significance level of 0.05.

## Results

### The retrieving time, the package opening time, and the donning time

The retrieving time, the package opening time, and the donning time were 5.8±4.2 seconds, 10.1±6.5 seconds, and 46.1±27.6 seconds, respectively. And 75[th] percentile of the retrieving time, the package opening time, and the donning time were 6.8 seconds, 12.8 seconds, and 57 seconds, respectively.

Table 4 showed the pass percentage in the package opening test and the donning test of different genders, different subjects, and different instruction conditions. Results showed that only 41.1% (62/151) of the subjects can open the package within 7 seconds, and only 21.2% (32/151) of study participants successfully donned a life preserver within 25 seconds. Both pass percentages lower than 75% required by TSO-C13g [9]. Male study participants had a higher pass percentage than female participants in the package opening test (47.8% of male students vs. 28.6% of female students, 61.1% of male villagers vs. 29.2% of female villagers) and the donning test (26.9% of male students vs. 16.7% of female students, 33.3% of male villagers vs. 4.2% of female villagers). The pass percentage under instruction condition I was significantly lower than that under instruction condition II (11.5% vs. 62.1%).

### The post-test questionnaire

The post-test questionnaire showed that 85.4% (129/151) of the subjects had difficulty in correctly donning the life preserver. The main obstacles included: straps, confusion on top/bottom or front/back, confusion on the hole, package problem, nervous or hesitant, retrieving problem, and confusion on the briefing card.

About 27.2% (41/151) of the subjects reported that the straps were too long for them to know how to use. They often fail to fasten the straps correctly and had to wait for the staff or the neighbor to remind them to tighten properly. About 21.9% (33/151) of the subjects were

**Table 5. T-test for potential influential factors with two-level categories.**

| Influential factor | Donning performance | T | Sig. | Mean differences | Standard error | 95% confidence intervals | |
|---|---|---|---|---|---|---|---|
| | | | | | | Lower | Upper |
| Gender | Retrieving time | -2.017 | .045* | -1.379 | 0.684 | -2.730 | -0.028 |
| | Package opening time | -3.284 | .001* | -3.629 | 1.105 | -5.822 | -1.436 |
| | Donning time | .076 | .939 | 0.346 | 4.550 | -8.644 | 9.337 |
| Urban/rural | Retrieving time | -.963 | .337 | -0.671 | 0.697 | -2.047 | 0.706 |
| | Package opening time | 1.885 | .063 | 2.200 | 1.167 | -0.118 | 4.517 |
| | Donning time | -.402 | .688 | -1.844 | 4.585 | -10.905 | 7.216 |
| Villager/student | Retrieving time | -.120 | .905 | -0.092 | 0.767 | -1.608 | 1.424 |
| | Package opening time | .967 | .335 | 1.149 | 1.189 | -1.200 | 3.498 |
| | Donning time | -1.105 | .271 | -5.541 | 5.016 | -15.453 | 4.371 |
| Wearing glasses | Retrieving time | -.498 | .619 | -0.342 | 0.687 | -1.699 | 1.016 |
| | Package opening time | -1.332 | .185 | -1.419 | 1.065 | -3.526 | 0.688 |
| | Donning time | .918 | .360 | 4.131 | 4.501 | -4.763 | 13.025 |
| Instruction condition | Retrieving time | .716 | .475 | 0.624 | 0.871 | -1.097 | 2.346 |
| | Package opening time | 1.459 | .147 | 1.965 | 1.347 | -0.696 | 4.626 |
| | Donning time | 10.242 | .000* | 27.847 | 2.719 | 22.473 | 33.220 |

Note:

* p value is significant at 0.05.

confused about what was the top/bottom or front/back so that they cannot don the life pre-server quickly. About 21.9% (33/151) of the subjects complained that the hole of the life pre-server was too small to see, and prevented participants with glasses to don it quickly and correctly. For the sake of unification, all packages containing life preservers were made on site. Compared with the original package, the package was made of the same material but easier to open. But 21.9% (33/151) of the subjects still complained that the package was hard to open. In the donning test process, about 18.5% (28/151) of the subjects were nervous or hesitant, because they did not know what to do when they saw the life preserver. About 7.9% (12/151) of the subjects complained that they had trouble in retrieving packages under the seat, and seat belts restrained them and prevented them from retrieving packages quickly. About 6.0% (9/151) of the subjects questioned that the pictures in the briefing card were different from the actual life preservers, and regarded the card misled them.

## T-test and ANOVA

T-test for potential influential factors with two-level categories was shown in Table 5. Results showed that gender had a significant effect on the retrieving time and the package opening time, while instruction condition was found to have a significant impact on the donning time.

ANOVA for potential influential factors with three or more categories was shown in Table 6. Results showed that age group had a significant effect on the donning time, but did not have a significant effect on the retrieving time and the package opening time. Results also showed that seat did not have a significant impact on the donning performance.

## The SLR

All variables were offered into the stepwise linear regression (SLR) to evaluate the influential factors for life preserver donning tests. Gender had a significant effect on the package opening

**Table 6. ANOVA for potential influential factors with three or more categories.**

| Influential factor | Donning performance | Comparison | Sum of squares | Mean square | F | Sig. |
|---|---|---|---|---|---|---|
| Seat | Retrieving time | Between Groups | 13.204 | 4.401 | .245 | .865 |
| | | Within Groups | 2645.604 | 17.997 | | |
| | | Total | 2658.808 | | | |
| | Package opening time | Between Groups | 46.535 | 15.512 | .358 | .784 |
| | | Within Groups | 6375.544 | 43.371 | | |
| | | Total | 6422.079 | | | |
| | Donning time | Between Groups | 776.807 | 258.936 | .334 | .800 |
| | | Within Groups | 113823.048 | 774.306 | | |
| | | Total | 114599.854 | | | |
| Age group | Retrieving time | Between Groups | 35.566 | 8.892 | .495 | .740 |
| | | Within Groups | 2623.242 | 17.967 | | |
| | | Total | 2658.808 | | | |
| | Package opening time | Between Groups | 203.994 | 50.999 | 1.197 | .314 |
| | | Within Groups | 6218.085 | 42.590 | | |
| | | Total | 6422.079 | | | |
| | Donning time | Between Groups | 9504.551 | 2376.138 | 3.301 | .013* |
| | | Within Groups | 105095.303 | 719.831 | | |
| | | Total | 114599.854 | | | |

Note:

* p value is significant at 0.05.

time, and instruction condition was found to have a significant impact on the donning time. SLR also showed that tool test time had a significant effect on the retrieving time, the package opening time, and the donning time, while self-test score did not have a significant effect. Self-test was self-evaluating, and may not fully represent the subject's flexibility due to the inconsistency of subjective cognition and evaluation standards. While tool test required the subject to perform hands-on tasks on site, which can better represent his or her flexibility.

The SLR models of the retrieving time, the package opening time, and the donning time took the following forms

$$y_{retrieving} = 0.019 + 0.023x_{14} \tag{1}$$

(RMSE = 4.084, $r^2$ = 0.059, F = 10.74, p = 0.002)

$$y_{opening} = -1.340 + 3.446x_1 + 0.026x_{14} \tag{2}$$

(RMSE = 6.214, $r^2$ = 0.098, F = 9.162, p = 0.000)

$$y_{donning} = 50.761 - 27.001x_7 + 0.111x_{14} \tag{3}$$

(RMSE = 25.005, $r^2$ = 0.182, F = 17.642, p = 0.000)

Where $y_{retrieving}$, $y_{opening}$, and $y_{donning}$ were independent variables about the retrieving time, the package opening time, and the donning time, respectively. $x_{14}$, $x_1$, and $x_7$ were the dependent variables about tool test time, gender, and instruction condition. $r^2$ in three equations were low means that the $x$ variables can explain a small part of the change in $y$. For example, $r^2$ in the Eq (1) is low means the tool test time can explain a small part change in the retrieving

time, but the retrieving time was not entirely dependent on the tool test time. Corbett et al. [6] also proved that $r^2$ of the participant age and the donning test time was low.

## Discussions

### The difficulty of donning the life preserver

Table 4 and the post-test questionnaire showed the difficulty of donning the life preserver. Corbett et al. [6] regarded that the major difficulty to correct donning was the straps and confusion on top/bottom or front/back. Rasmussen and Steen [7] pointed out that the straps were the major obstacle. Package problem and retrieving problem for donning the life preserver both have a long history [1]. Passengers occasionally find difficulties in retrieving life preservers from under the seat [6,7,13], especially in low lighting and cold weather [14].

Obstacles not mentioned in previous studies included: confusion on the hole, nervous or hesitant, and confusion on the briefing card. Many subjects could not find the hole in the life preserver for a long time. The main reason was that there was no connection between the outer edges of two chambers, which made the participants initially thought that the hole was between the two chambers. Some subjects with glasses complained that the hole was too small for them. But the t-test results showed that wearing glasses did not have a significant effect on the donning performance, and gender also did not have a significant effect on the donning time. The problems of "nervous or hesitant" and "confusion on the briefing card" showed the subjects were not familiar with life preservers, the main reason was that the instruction was not enough, and the briefing cards were not accurate and detailed for them.

### The retrieving test

Retrieving time is one of the most important parts of the donning performance of the life preserver, so the retrieving test should be included in TSO-C13 series standard. The retrieving time in this paper was 5.8±4.2 seconds, and the 75th percentile of retrieving time was 6.8 seconds. About 16.6% (25/151) of the subjects cannot retrieve a life preserver from under the seat within 7 seconds, which showed the troubles that some participants experienced with retrieving life preservers. FAA tests confirmed that many passengers may take at least 7 to 8 seconds to retrieve a life preserver [6,7]. Gowdy and DeWeese [13] investigated retrieving life preservers from under the seat in 2003. The mean retrieving time of four configurations was 7.4 seconds, 8.5 seconds, 13.3 seconds, and 15.3 seconds, respectively, which meant that some configurations cannot be considered as easy to retrieve life preservers. Based on the above analysis, it is recommended that 75% of the total number of test participants must complete retrieving the life preserver within 7 seconds in the retrieving test.

### The factors related to the donning test in the standards

**Age.** Table 5 showed that age group had a significant effect on the donning time, but did not have a significant effect on the retrieving time and the package opening time. Corbett et al. [6] proved that age was correlated with the donning time ($r^2$ = 0.0841, p <0.01). But they also regarded that age was correlated with the package opening time ($r^2$ = 0.0324, p = 0.03), which was different from the finding of this paper. The reason was that the participants used in their research ranged from 23 to 75 years, while the participants in this test were all under 60 years old. Females over the age of 60 may take more time to open the package [6]. Runnarong et al. [15] pointed out that reach-to-grasp performance deteriorated with age. In the vibrotactile display test by Bao et al. [16], average reaction time for old adults was 60 ms slower than that for young adults.

To further assess the effect of different age groups on the donning time, the post hoc multiple comparisons of one-way ANOVA by the Least Significant Difference (LSD) test were used, see S1 Table. The age group of "20–29" was significantly different from other groups except for the group of "30–39", and the differences between other groups were not significant.

Thus, although age was not included in the SLR model, it is reasonable to group ages in the donning test due to partly significant differences between age groups.

**Gender.** In the donning tests of this paper, the retrieving time, the package opening time, and the donning time of the male subjects were 5.2±3.3 seconds, 8.5±4.6 seconds, and 46.3 ±30.8 seconds, respectively. While those of female subjects were 6.6±5.0 seconds, 12.2±8.0 seconds, and 45.9±23.2 seconds, respectively. The retrieving time and the package opening time of male subjects were shorter than those of female subjects. Table 4 showed male study participants had a higher pass percentage than female participants in the package opening test and the donning test. Table 5 proved that gender had a significant effect on the retrieving time and the package opening time, and gender was included in the SLR model of the package opening time. Males typically have greater absolute levels of muscle size and strength than females [17]. Corbett et al. [6] proposed older females may have difficulty in opening the package. Their evidence was that an older woman spent 37.9 seconds in opening her life preserver from the package. Sialino et al. [18] also pointed out that older women perform consistently poorer on physical performance tests compared to men.

Gender had a significant effect on the package opening time, and older female subjects were relatively slower in opening packages, so it is reasonable that females over the age of 60 were chosen as subjects in the package opening test [9]. For the same reason, if the retrieving test was added in the revised TSO-C13 series standard, females over the age of 60 also should be chosen as subjects in the retrieving test.

Since gender had no significant effect on the donning time, the gender requirement, i.e., the same sex should not exceed 60% of the total number of test subjects, should be deleted in the donning test.

**Test subject characteristics.** The SLR models showed that height, weight, and head circumference did not have a significant effect on the retrieving time, the package opening time, and the donning time.

Pearson correlation analysis was used to assess the strength of the relationship between test subject characteristics and the donning performance, see S2 Table. The Pearson correlation coefficients were all very small (<0.4) and belonged to weak correlation or irrelevant. Pearson correlation also showed that test subject characteristics did not have a significant effect on the retrieving time, the package opening time, and the donning time, which coincided with the t-test results.

## Conclusions and recommendations

This study once again proved the difficulty of retrieving life preservers, opening packages, and donning life preservers. Four of fourteen variables, including gender, instruction condition, age group, and tool test time, were identified as influencing factors for life preserver donning performance. Recommendations for modification of donning test procedure are as follows.

1. Retrieving time is one of the most important parts of the donning performance, so the retrieving test should be included in TSO-C13g. Considering the subjects required by package opening test in TSO-C13g, it is recommended that retrieving the life preserver should be demonstrated with 7 seconds by 8 of 10 females the age of 60, without a preview of instructions. In cases for which additional participants are required, 75% of the total number of test participants must complete retrieving the life preserver within 7 seconds.

2. The subject's flexibility had a significant effect on the retrieving time, the opening time, and the donning time. Thus, there should be a certain percentage of test participants with different flexibility. It is recommended that tests of the subject's flexibility such as the GATB should be conducted when selecting subjects, and the subjects with excellent, general and poor flexibility should be approximately equal.

3. Since gender had no significant effect on the donning time, the gender requirement, i.e., the same sex should not exceed 60% of the total number of test subjects, should be deleted in the donning test. That is, as long as ensuring 10 females the age of 60 for the retrieving test and the package opening test, there is no gender requirement for the donning performance test when selecting subjects.

4. Test subject characteristics such as height, weight, and head circumference did not have a significant effect on the donning performance, thus, the subject characteristics defined in TSO-C13g should be removed.

5. Since the instruction condition had a significant effect on the life preserver donning test, different donning demonstrations in the donning test should correspond to different donning time requirements. There were three levels of instruction condition in TSO-C13g: no donning instruction, a typical preflight video briefing, and donning demonstration. According to this study, it is recommended that at least 75% of the total number of test subjects, and at least 60% of the test subjects in each age group, can don the life preserver within 25 seconds under donning demonstration, within 40 seconds under a typical preflight video briefing and within 50 seconds under no donning instruction.

6. Life preservers and briefing cards should be optimized to ensure a better donning performance. The optimization of life preservers includes connecting outer edges of two chambers to avoid being misunderstood, color-coded straps instead of traditional straps, new type with the easy donning performance such as "vest" life preserver. The optimization of briefing cards includes lively colors, forms that exactly matches the life preserver aboard airplanes, separated briefing cards for adult life preserver.

This paper only studied a typical standard life preserver and two rows of air carrier coach class triple-seat. The participants were also limited to college students and villagers. The next research should expand more research objects to further verify the validity of the conclusions in this paper.

## Supporting information

**S1 Table. The post hoc multiple comparisons of one-way ANOVA by LSD test.**
(DOCX)

**S2 Table. Pearson correlation between test subject characteristics and donning performance.**
(DOCX)

**S1 File. The questionnaire and tool tests.**
(DOCX)

**S2 File. Detailed experiment data file.**
(XLSX)

## Acknowledgments

We thank Miss Jiehuan Lu for help in the experiments.

## Author Contributions

**Conceptualization:** Ruiliang Yang.

**Data curation:** Zijiang Wu.

**Formal analysis:** Zijiang Wu.

**Funding acquisition:** Ruiliang Yang.

**Investigation:** Ruiliang Yang.

**Methodology:** Zijiang Wu.

**Project administration:** Xiaoming Qian.

**Resources:** Xiaoming Qian.

**Software:** Xiaoming Qian.

**Supervision:** Xiaoming Qian.

**Validation:** Xiaoming Qian.

**Writing – original draft:** Ruiliang Yang.

**Writing – review & editing:** Zijiang Wu.

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
