## [Decision Letter · Decision Letter 0]

29 Dec 2020

PONE-D-20-34428

Evaluating the influential factors for life preserver donning tests

PLOS ONE

Dear Dr. Qian,

Thank you for submitting your manuscript to PLOS ONE. After careful consideration, we feel that it has merit but does not fully meet PLOS ONE’s publication criteria as it currently stands. Therefore, we invite you to submit a revised version of the manuscript that addresses the points raised during the review process.

Please, make all the corrections required by the reviewers.

We look forward to receiving your revised manuscript.

Kind regards,

Ahmed Mancy Mosa, Ph.D.

Academic Editor

PLOS ONE

2.Please provide additional details regarding participant consent. In the ethics statement in the Methods and online submission information, please ensure that you have specified (1) whether consent was informed and (2) what type you obtained (for instance, written or verbal, and if verbal, how it was documented and witnessed). If your study included minors, state whether you obtained consent from parents or guardians. If the need for consent was waived by the ethics committee, please include this information.

4.We note that Figure 2 and 3  includes an image of a patient / participant in the study. 

Please respond by return e-mail with an amended manuscript. We can upload this to your submission on your behalf.

If you are unable to obtain consent from the subject of the photograph, please either instruct us to remove the figure or supply a replacement figure by return e-mail for which you hold the relevant copyright permissions and subject consents. In some cases, you may need to specify in the text that the image used in the figure is not the original image used in the study, but a similar image used for illustrative purposes only. We can make any changes on your behalf.

Journal Requirements:

Additional Editor Comments (if provided):

Please, consider all the comments of the reviewers carefully

Reviewers' comments:

Reviewer's Responses to Questions

**Comments to the Author**

1. Is the manuscript technically sound, and do the data support the conclusions?

Reviewer #1: Yes

Reviewer #2: Partly

Reviewer #3: Yes

Reviewer #4: Partly

2. Has the statistical analysis been performed appropriately and rigorously? 

Reviewer #1: Yes

Reviewer #2: N/A

Reviewer #3: Yes

Reviewer #4: Yes

3. Have the authors made all data underlying the findings in their manuscript fully available?

Reviewer #1: Yes

Reviewer #2: No

Reviewer #3: Yes

Reviewer #4: Yes

4. Is the manuscript presented in an intelligible fashion and written in standard English?

Reviewer #1: Yes

Reviewer #2: No

Reviewer #3: Yes

Reviewer #4: Yes

5. Review Comments to the Author

Reviewer #1: As we all know, Life preservers usually play a vital role in ensuring the safety of passengers. In order to evaluate the influencing factors of the Life preservers donning test, the author selected 109 college students and 42 villagers as the research objects. A total of fourteen variables are considered as potential influencing factors. T-test or one-way analysis of variance is used to determine whether the grouping of categorical variables has a significant impact on donning performance. In order to evaluate the relationship between donning performance and influencing factors, a model of retrieving time, opening time and donning time was established based on SLR analysis. Finally, a suggestion to modify the donning test program is introduced, which helps to improve the effectiveness and reliability of the donning test. The structure of the research content of the thesis is complete, the experiment is reasonable, and the data analysis is relatively complete. However, there are certain problems in the paper, and the overall quality of the paper can be improved after revision. Therefore, it is recommended a major revision. The specific issues are as follows:

1、There are several grammatical issues in the paper, I suggest that the author read through the full manuscript and carefully modify the deficiencies.

2、It is unclear that the data analysis and process techniques mentioned by the authors are their own proposed work or are they addressing existing work. Authors should clearly specify that and provide citation and reference to the existing work.

3、The quality of figures and their placement needs to be improved. At times, it becomes difficult the link the discussion with figures in the article. Lastly, the authors are requested to highlight their technical contribution more precisely and how it aligns with the publication criteria set by PLOS ONE.

4、Authors should provide discussions on their analysis results and they should also compare their results with existing and related studies.

5、The sample researched in the thesis seems to be a bit small. Appropriately increasing the sample size will make the research results more convincing; in addition, before the accident, passengers are generally out of psychological stress reaction stage, the ability to operate equipment will be reduced, and the data calculation process Should this part of the content be considered?

Reviewer #2: It is a messy article without suitable coherence.

The subject characteristics must be presented in the materials and method section.

All tables must be presented at result section.

Some tables, such as Table 7, are too long and is not suitable by this way for an article.

The discussion section should be rewritten based on the essay writing framework.

The article needs a general review.

Reviewer #3: I appreciate authors for conducting an interesting study to evaluate the influencing factor for life preserver donning tests.

Title of the study is Evaluating the influential factors for life preserver donning tests and objective of the study is to evaluate the influential factors for donning tests of adult life preservers and propose recommendations for modification of the donning test procedure.

An appropriate study design was selected, and the procedure described clearly. Results are presented properly. The conclusion was made according to the study result.

Well done!

Reviewer #4: The study “Evaluating the influential factors for life preserver donning tests” is interesting. This paper aims to evaluate the influential factors for donning tests of adult life preservers and propose recommendations for modification of the donning test procedure. The paper is well set, and the problem highlighted executed properly. However, attention should be given to the following highlighted points before resubmitting.

1. Line 26 T-test or ANOVA may be written as “T-test and ANOVA”

2. The character shorthand should defined on their first appearance and after that use accordingly. Line 48 TSO-C13, Line 49 FAA etc. these abbreviations is not defined. Or make a table which include all character shorthand. See for reference https://www.hindawi.com/journals/mpe/2020/1325071/

3. Table 1, Test subject characteristics (Head circumference (cm) 5th, 50th , and 95th for male, the 5th and 50th percentile are Nil while for female the 50th and 95th percentiles are Nile. Please elaborate this point that how come a specific percentile is Nil when 95th is available and 50th is Nil.

4. Lines 149 Check this percentage “About 56.3% of the participants were male, and 41.1% were female. Why the sum of two percentages is not 100 when the data are categorial having two categories i.e. Male and Female.

5. Lines 236 “The step stepwise linear regression (SLR) models of the retrieving time, the package opening time, and the donning time took the following forms Eq (1), Eq (2), and Eq (3). Where y retrieving, y opening, and y donning were the independent variables about the retrieving time, the package opening time, and the donning time, respectively. x14, x1, and x7 were the dependent variables about tool test time, gender, and instruction condition.” All the three models are significant at 5% level of significance but their R2 is very low, for Eq (1) it is 0.059, for Eq (2) is 0.098 and for Eq (3) is 0.182. Is there any specific reason that why R2 is low.

6. Lines 281 “Some subjects with glasses and some girl with long hair complained that the hole was too small for them. But the t test results showed that wearing glasses did not have a significant effect on the donning performance, and gender also did not have a significant effect on the donning time.” Here girl with long hair is not tested also test this variable as well.

7. In last more recent references should be added to broaden the view of readers and enhance the new contribution of this paper for comparison.

6. PLOS authors have the option to publish the peer review history of their article (what does this mean?). If published, this will include your full peer review and any attached files.

Reviewer #1: No

Reviewer #2: No

Reviewer #3: **Yes: **Saidur Mashreky

Reviewer #4: No

---

## [Author Response · Author response to Decision Letter 0]

13 Jan 2021

We would like to express our sincere thanks to the reviewers for the constructive and positive comments.

Replies to Anita Estes

As there are potentially identifying images in your manuscript, please confirm whether the participants also explicitly provided their consent for their images to be published.

There are 7 people in the revised manuscript, and they all signed the consent form for publication in PLOS ONE. 

The following sentence was added to the revised manuscript.

The participants in the figures have given written informed consent to publish their images.

Replies to editor

1. Please ensure that your manuscript meets PLOS ONE's style requirements, including those for file naming ?

Ok. The revised manuscript meets PLOS ONE's style requirements.

2.Please provide additional details regarding participant consent. In the ethics statement in the Methods and online submission information, please ensure that you have specified (1) whether consent was informed and (2) what type you obtained (for instance, written or verbal, and if verbal, how it was documented and witnessed). If your study included minors, state whether you obtained consent from parents or guardians. If the need for consent was waived by the ethics committee, please include this information.

The study was approved by Human Research Ethics Committee for Non-Clinical Faculties of School of Mechanical Engineering, Tiangong University. All participants were verbally informed of the contents of the experiment, then signed their names to agree the experiment.

The ethics statement in the Methods and online submission information have specified the above information.

The study did not report any medical records or archived samples.

Ok. Captions for the Supporting Information files were included at the end of the revised manuscript, and in-text citations were updated to match accordingly.

4.We note that Figure 2 and 3 includes an image of a patient / participant in the study. As per the PLOS ONE policy (http://journals.plos.org/plosone/s/submission-guidelines#loc-human-subjects-research) on papers that include identifying, or potentially identifying, information, the individual(s) or parent(s)/guardian(s) must be informed of the terms of the PLOS open-access (CC-BY) license and provide specific permission for publication of these details under the terms of this license. Please download the Consent Form for Publication in a PLOS Journal (http://journals.plos.org/plosone/s/file?id=8ce6/plos-consent-form-english.pdf). The signed consent form should not be submitted with the manuscript, but should be securely filed in the individual's case notes. Please amend the methods section and ethics statement of the manuscript to explicitly state that the patient/participant has provided consent for publication: “The individual in this manuscript has given written informed consent (as outlined in PLOS consent form) to publish these case details”.

There are 7 people in Figures 2 and 3, and they all signed the consent form for publication in PLOS ONE.

Replies to Reviewer #1

1. There are several grammatical issues in the paper, I suggest that the author read through the full manuscript and carefully modify the deficiencies.

 Thanks for the suggestion of the reviewer. 

 The authors read through the full manuscript and carefully modify some deficiencies. Then the manuscript was grammatically checked by the word grammar checker and Ginger, and then checked by a native English teacher.

2.It is unclear that the data analysis and process techniques mentioned by the authors are their own proposed work or are they addressing existing work. Authors should clearly specify that and provide citation and reference to the existing work.

 Thanks for the suggestion of the reviewer. 

 The data analysis and process techniques in the manuscript are our own proposed work, and the statistical analysis used in this manuscript are very mature and reliable. 

 In the section of “discussion”, citation and reference to the existing work were specified to compare with this manuscript.

3.The quality of figures and their placement needs to be improved. At times, it becomes difficult the link the discussion with figures in the article. Lastly, the authors are requested to highlight their technical contribution more precisely and how it aligns with the publication criteria set by PLOS ONE.

 Thanks for the suggestion of the reviewer. 

 In the original manuscript, Fig 3 included four small pictures, which may cause confusion. For more clarity, there was only one picture in Fig 3 in the revised manuscript.

 The quality of figures in the revised manuscript all met the “Figure File Requirements” of the journal. The figure captions were inserted immediately after the first paragraph in which the figure is cited.

 “Data Curation”, “Formal Analysis”, ”Funding Acquisition”, ” Investigation”, “Project Administration”, “Software” and “Resources” were added in the “Author Contributions” section, which highlighted the authors’ technical contribution more precisely.

4、Authors should provide discussions on their analysis results and they should also compare their results with existing and related studies.

 Thanks for the suggestion of the reviewer. 

 Discussions on the analysis results and comparison of the results and related studies were provided in the revised manuscript.

(1) Three papers about the donning performance of life preservers were discussed and compared in the “Discussion” section.

 [1] Corbett CL, Weed DB, Ruppel DJ, Larcher KG, McLean GA. Inflatable emergency equipment I: evaluation of individual inflatable aviation life preserver donning tests. Technical Report of the U.S. Federal Aviation Administration. Report no. DOT/FAA/AM-14/14. December 2014. Available from: https://www.faa.gov/data_research/research/med_humanfacs/oamtechreports/2010s/media/201414.pdf.

 [2] Rasmussen PG, Steen J. Retrieval and donning of inflatable life preservers. Technical Report of the U.S. Federal Aviation Administration. Report No: AAC-119-83-5, July 1983. 

 [3] Gowdy V, DeWeese R. Human factors associated with the certification of airplane passenger seats: life preserver retrieval. Technical Report of the U.S. Federal Aviation Administration. Report no. DOT/FAA/AM-03/9, May 2003. Available from: https:// www.faa.gov/data_research/research/med_humanfacs/oamtechreports/2000s/media/0309.pdf.

 (2) Four papers about the sex differences or age differences were discussed and compared in the “Discussion” section.

 [1] Runnarong N, Tretriluxana J, Waiyasil W, Sittisupapong P, Tretriluxana S. Age-related changes in reach-to-grasp movements with partial visual occlusion. PLoS ONE 2019; 14(8): e0221320. doi: 10.1371/journal.pone.0221320.

 [2]Bao T, Su L, Kinnaird C, Kabeto M, Shull PB, Sienko KH (2019) Vibrotactile display design: Quantifying the importance of age and various factors on reaction times. PLoS ONE 2019; 14(8): e0219737. doi: 10.1371/journal.pone.0219737.

 [3] Jones MD, Wewege MA, Hackett DA, Keogh JWL, Hagstrom AD. Sex differences in adaptations in muscle strength and size following resistance training in older adults: A systematic review and meta-analysis. Sports Med. 2020. doi:10.1007/s40279-020-01388-4

 [4]Sialino LD, Schaap LA, van Oostrom SH, Nooyens ACJ, Picavet HSJ, Twisk JWR, et al. Sex differences in physical performance by age, educational level, ethnic groups and birth cohort: The Longitudinal Aging Study Amsterdam. PLoS ONE 2019; 14(12): e0226342. doi: 10.1371/journal.pone.0226342.

5. The sample researched in the thesis seems to be a bit small. Appropriately increasing the sample size will make the research results more convincing; in addition, before the accident, passengers are generally out of psychological stress reaction stage, the ability to operate equipment will be reduced, and the data calculation process Should this part of the content be considered?

 Thanks for the suggestion of the reviewer. 

 (1)According to TSO-C13g [1], at least 25 test subjects shall be employed in donning tests of an adult preserver. Furthermore, there were 6 people in each type donning test by the FAA report in 2014 [2]. 

 In this manuscript, a total of 151 subjects were used in the donning test, which meets the sample requirements of the TSO-C13g. And the sample size was much more than that of the FAA report [2].

 Of course, more sample size will make the research results more convincing, so the authors will use more samples in subsequent research.

[1] TSO-C13g. Life preservers, Technical Standard Order of Federal Aviation Administration. Issued date: February 3, 2017.

[2] Corbett CL, Weed DB, Ruppel DJ, Larcher KG, McLean GA. Inflatable emergency equipment I: evaluation of individual inflatable aviation life preserver donning tests. Technical Report of the U.S. Federal Aviation Administration. Report no. DOT/FAA/AM-14/14. December 2014. Available from: https://www.faa.gov/data_research/research/med_humanfacs/oamtechreports/2010s/media/201414.pdf.

(2) The donning test of TSO-C13g does not consider the psychological stress by the accident, because the donning test was carried out in the laboratory. The manuscript aims to evaluate influencing factors for life preserver donning tests, so the psychological stress by the accident was not considered in the manuscript. 

Of course, psychological stress exists in the water-related accident, but difficult to be quantified in the laboratory. The authors will consider this content in subsequent research.

Replies to Reviewer #2

1. The subject characteristics must be presented in the materials and method section.

 Ok. All subject characteristics were presented in the method section.

2.All tables must be presented at result section.

Ok. Tables 7 and 8 were moved to Supporting Information, and all tables were presented in the result section in the revised manuscript. 

3. Some tables, such as Table 7, are too long and is not suitable by this way for an article.

Ok. Table 7 was move to Supporting Information in the revised manuscript. 

4. The discussion section should be rewritten based on the essay writing framework.

The discussion section was rewritten based on the essay writing framework. 

（1）“The post-test questionnaire” was moved from the discussion section to the result section.

（2）“The retrieving test” was moved from the discussion section to the result section.

（3）Tables 7 and 8 were moved to Supporting Information in the revised manuscript. 

 （4）Four papers were added in the discussion section for comparison, as follows:

[1] Runnarong N, Tretriluxana J, Waiyasil W, Sittisupapong P, Tretriluxana S. Age-related changes in reach-to-grasp movements with partial visual occlusion. PLoS ONE 2019; 14(8): e0221320. doi: 10.1371/journal.pone.0221320.

[2]Bao T, Su L, Kinnaird C, Kabeto M, Shull PB, Sienko KH (2019) Vibrotactile display design: Quantifying the importance of age and various factors on reaction times. PLoS ONE 2019; 14(8): e0219737. doi: 10.1371/journal.pone.0219737.

[3] Jones MD, Wewege MA, Hackett DA, Keogh JWL, Hagstrom AD. Sex differences in adaptations in muscle strength and size following resistance training in older adults: A systematic review and meta-analysis. Sports Med. 2020. doi:10.1007/s40279-020-01388-4

[4]Sialino LD, Schaap LA, van Oostrom SH, Nooyens ACJ, Picavet HSJ, Twisk JWR, et al. Sex differences in physical performance by age, educational level, ethnic groups and birth cohort: The Longitudinal Aging Study Amsterdam. PLoS ONE 2019; 14(12): e0226342. doi: 10.1371/journal.pone.0226342.

5. The article needs a general review.

The authors read through the full manuscript and carefully modify some deficiencies. Then the manuscript was grammatically checked by the word grammar checker and Ginger, and then checked by a native English teacher.

Replies to Reviewer #3

We would like to express our sincere thanks to the reviewer #3 for the constructive and positive comments.

Replies to Reviewer #4

1. Line 26 T-test or ANOVA may be written as “T-test and ANOVA”

Ok. “T-test or ANOVA” was changed to “T-test and ANOVA” in the revised manuscript.

2. The character shorthand should defined on their first appearance and after that use accordingly. Line 48 TSO-C13, Line 49 FAA etc. these abbreviations is not defined. Or make a table which include all character shorthand. See for reference https://www.hindawi.com/journals/mpe/2020/1325071/

 Thanks for the suggestion of the reviewer. “TSO-C13” was changed to “Technical Standard Order (TSO)”, and “FAA” was changed to “Federal Aviation Administration (FAA)” in the revised manuscript.

3. Table 1, Test subject characteristics (Head circumference (cm) 5th, 50th , and 95th for male, the 5th and 50th percentile are Nil while for female the 50th and 95th percentiles are Nile. Please elaborate this point that how come a specific percentile is Nil when 95th is available and 50th is Nil.

 “Nil” means “no requirement” in the TSO-C13g. According to TSO-C13g，test subjects are nominally defined as follows:

A 5th percentile male is no more than 1.7 m tall and weighs no more than 63.9 kg.

A 50th percentile female is 1.6 m tall and weighs 64.4 to 70.7 kg.

A 50th percentile male is 1.8 m tall and weighs 79.4 to 85.6 kg.

A 95th percentile female is at least 1.7 m tall and weighs at least93.0 kg.

A 95th percentile male is at least 1.9 m tall and weighs at least 110.7 kg with a head circumference of at least 60.4 cm.

 The original form may be misunderstood, so in the revised manuscript it was revised to a more suitable form.

4. Lines 149 Check this percentage “About 56.3% of the participants were male, and 41.1% were female. Why the sum of two percentages is not 100 when the data are categorial having two categories i.e. Male and Female.

 Thanks for the suggestion of the reviewer. This is a typographical error and has been corrected in the revised manuscript. The proportion of female in the table is 43.7%. This sentence is rewritten as:

 About 56.3% of the participants were male, and 43.7% were female. 

5. Lines 236 “The step stepwise linear regression (SLR) models of the retrieving time, the package opening time, and the donning time took the following forms Eq (1), Eq (2), and Eq (3). Where y retrieving, y opening, and y donning were the independent variables about the retrieving time, the package opening time, and the donning time, respectively. x14, x1, and x7 were the dependent variables about tool test time, gender, and instruction condition.” All the three models are significant at 5% level of significance but their R2 is very low, for Eq (1) it is 0.059, for Eq (2) is 0.098 and for Eq (3) is 0.182. Is there any specific reason that why R2 is low.

(1) r2 is low means that the x variable can explain only part of the change in y, and other variables may be added to explain the change in y. For example, r2 in the Eq. (1) is low means the tool test time can explain only part change in the retrieving time, but the retrieving time was not entirely dependent on the tool test time. 

(2) Corbett et al. [1] also proved that age was correlated with the donning time (r2 = 0.0841, p <0.01). and the package opening time (r2 =0.0324, p=0.03). (Page 14 of Ref.[1]) 

[1] Corbett CL, Weed DB, Ruppel DJ, Larcher KG, McLean GA. Inflatable emergency equipment I: evaluation of individual inflatable aviation life preserver donning tests. Technical Report of the U.S. Federal Aviation Administration. Report no. DOT/FAA/AM-14/14. December 2014. Available from: https://www.faa.gov/data_research/research/med_humanfacs/oamtechreports/2010s/media/201414.pdf.

(3) The following paragraph was added in the revised manuscript.

r2 in three equations were low means that the x variables can explain a small part of the change in y. For example, r2 in Eq. (1) is low means the tool test time can explain a small part change in the retrieving time, but the retrieving time was not entirely dependent on the tool test time. Corbett et al. [6] also proved that r2 of the participant age and the donning test time was low.

6. Lines 281 “Some subjects with glasses and some girl with long hair complained that the hole was too small for them. But the t test results showed that wearing glasses did not have a significant effect on the donning performance, and gender also did not have a significant effect on the donning time.” Here girl with long hair is not tested also test this variable as well.

 Because long hair is difficult to define, it is impossible to accurately distinguish long hair, medium hair and short hair. Therefore, long hair is not used as a variable in the manuscript.

The two sentences were rewritten as:

Some subjects with glasses that the hole was too small for them. But the t test results showed that wearing glasses did not have a significant effect on the donning performance, and gender also did not have a significant effect on the donning time.”

7. In last more recent references should be added to broaden the view of readers and enhance the new contribution of this paper for comparison.

Thanks for the suggestion of the reviewer. 

Five papers were added in the revised manuscript for comparison, as follows:

[1] MacDonald CV, Brooks CJ , Kozey JW. Infant life jacket donning trials using children and their parents: Comparison to the Canadian standard. Int. J. Ind. Ergonom. 2016; 54:19-25. doi: 10.1016/j.ergon.2015.12.003

[2] Runnarong N, Tretriluxana J, Waiyasil W, Sittisupapong P, Tretriluxana S. Age-related changes in reach-to-grasp movements with partial visual occlusion. PLoS ONE 2019; 14(8): e0221320. doi: 10.1371/journal.pone.0221320.

[3] Bao T, Su L, Kinnaird C, Kabeto M, Shull PB, Sienko KH (2019) Vibrotactile display design: Quantifying the importance of age and various factors on reaction times. PLoS ONE 2019; 14(8): e0219737. doi: 10.1371/journal.pone.0219737.

[4] Jones MD, Wewege MA, Hackett DA, Keogh JWL, Hagstrom AD. Sex differences in adaptations in muscle strength and size following resistance training in older adults: A systematic review and meta-analysis. Sports Med. 2020. doi:10.1007/s40279-020-01388-4

[5]Sialino LD, Schaap LA, van Oostrom SH, Nooyens ACJ, Picavet HSJ, Twisk JWR, et al. Sex differences in physical performance by age, educational level, ethnic groups and birth cohort: The Longitudinal Aging Study Amsterdam. PLoS ONE 2019; 14(12): e0226342. doi: 10.1371/journal.pone.0226342.

---

## [Decision Letter · Decision Letter 1]

25 Jan 2021

Evaluating the influential factors for life preserver donning tests

PONE-D-20-34428R1

Dear Dr. Qian,

We’re pleased to inform you that your manuscript has been judged scientifically suitable for publication and will be formally accepted for publication once it meets all outstanding technical requirements.

Kind regards,

Ahmed Mancy Mosa, Ph.D.

Academic Editor

PLOS ONE

Additional Editor Comments (optional):

Reviewers' comments:

Reviewer's Responses to Questions

**Comments to the Author**

1. If the authors have adequately addressed your comments raised in a previous round of review and you feel that this manuscript is now acceptable for publication, you may indicate that here to bypass the “Comments to the Author” section, enter your conflict of interest statement in the “Confidential to Editor” section, and submit your "Accept" recommendation.

Reviewer #1: All comments have been addressed

Reviewer #2: (No Response)

Reviewer #4: (No Response)

2. Is the manuscript technically sound, and do the data support the conclusions?

Reviewer #1: Yes

Reviewer #2: No

Reviewer #4: (No Response)

3. Has the statistical analysis been performed appropriately and rigorously? 

Reviewer #1: Yes

Reviewer #2: No

Reviewer #4: (No Response)

4. Have the authors made all data underlying the findings in their manuscript fully available?

Reviewer #1: Yes

Reviewer #2: No

Reviewer #4: (No Response)

5. Is the manuscript presented in an intelligible fashion and written in standard English?

Reviewer #1: Yes

Reviewer #2: No

Reviewer #4: (No Response)

6. Review Comments to the Author

Reviewer #1: The structure of the research content of the thesis is complete, the experiment is reasonable, and the data analysis is relatively complete. The author has revised all the questions mentioned and it is recommended to accept.

Reviewer #2: My recommendations to article improvement have not been applied and the article is not satisfactory by this way.

Reviewer #4: (No Response)

7. PLOS authors have the option to publish the peer review history of their article (what does this mean?). If published, this will include your full peer review and any attached files.

Reviewer #1: No

Reviewer #2: No

Reviewer #4: No

---

## [Editor Report · Acceptance letter]

27 Jan 2021

PONE-D-20-34428R1 

Evaluating the influential factors for life preserver donning tests 

Dear Dr. Qian:

I'm pleased to inform you that your manuscript has been deemed suitable for publication in PLOS ONE. Congratulations! Your manuscript is now with our production department. 

Kind regards, 

on behalf of

Dr. Ahmed Mancy Mosa 

Academic Editor

PLOS ONE